# ROBO-INSTRUCT: SIMULATOR-AUGMENTED INSTRUCTION ALIGNMENT FOR FINETUNING CODE LLMS

## ABSTRACT

Open-weight LLMs are particularly appealing choices to generate training data for fine-tuning Code LLMs on domain-specific service robot applications because they are cost-effective, customizable, and offer better privacy protection. However, unlike proprietary LLMs, open-weight models are more error-prone and often produce programs that violate domain-specific constraints. A promising solution is to incorporate a robot simulator with a well-defined environment to verify program correctness. Yet, these environments require pre-enumeration of relevant entities and their states, which limits the diversity of programs that can be effectively verified. In this work, we introduce ROBO-INSTRUCT that preserves the diversity of programs generated by an LLM while providing the correctness of simulator-based checking. ROBO-INSTRUCT introduces ROBOSIM to *dynamically synthesize consistent simulation environments* for each generated program. Moreover, ROBO-INSTRUCT handles subtler instruction-program inconsistencies that do not result in a constraint violation via INSTALIGN, an LLM-aided instruction-program alignment process. Given domain-specific APIs and a few seed examples, ROBO-INSTRUCT can leverage an 8B Llama3 model to generate a training dataset for fine-tuning a 7B CodeLlama model. Our fine-tuned model achieves a 28.75% improvement in pass@1 over the original base model and a 13.75% improvement compared to its SELF-INSTRUCT-finetuned counterparts, even surpassing the performance of a few proprietary LLMs, such as GPT-3.5-Turbo and Gemini-Pro.

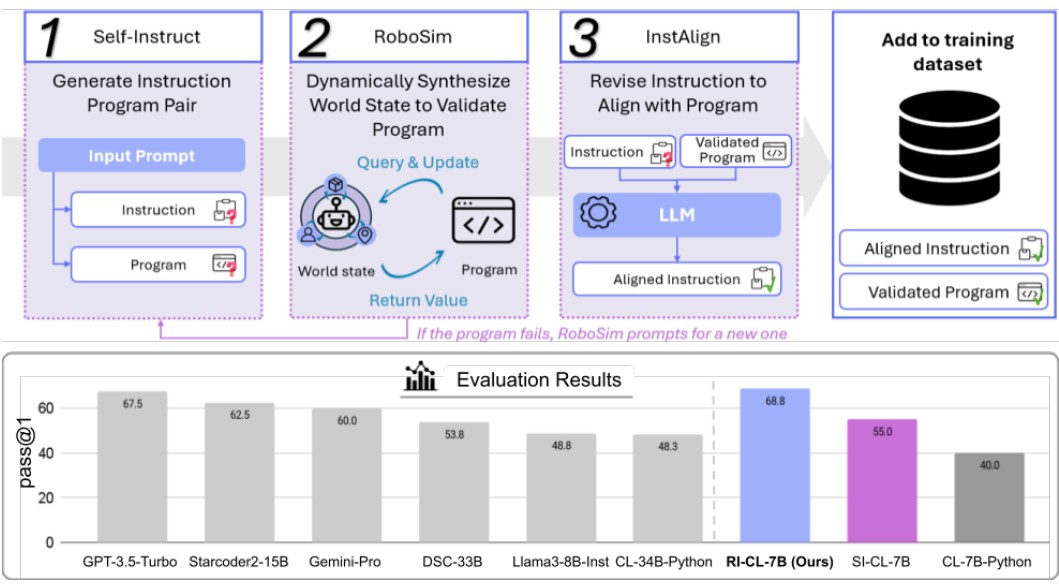

Figure 1: High-level overview of the ROBO-INSTRUCT framwork. This figure also shows the pass@1 score performance of ROBO-INSTRUCT fine-tuned LLM compared to other LLMs on ROBOEVAL.

# 1 INTRODUCTION

Large language models (LLMs) have shown great promise at leveraging domain-specific application programming interfaces (APIs) and generating robot programs from natural language instructions (Hu et al., 2024; Huang et al., 2023b; Biggie et al., 2023; Liu et al., 2023a; Wu et al., 2023; Liang et al., 2022; Singh et al., 2023; Huang et al., 2023a). For instance, by formulating a robot's navigation and perception skills into APIs, such as `go_to(location)` and `is_in_room(object)`, an LLM can generate a program for a service mobile robot to complete the task: *"Determine the number of conference rooms without markers"*. However, despite impressive results, the performance gap remains wide between proprietary and open-weight LLMs in generating robot programs from domain-specific APIs (Hu et al., 2024). To bridge the performance gap, SELF-INSTRUCT (Wang et al., 2022) is a popular method for generating domain-specific data for finetuning LLMs. Given the definition of APIs and a few seed task examples, SELF-INSTRUCT prompts an LLM to generate diverse instruction-program pairs as training data.

This makes SELF-INSTRUCT a seemingly appealing way to prompt open-weight LLMs to generate training data, for the fine-tuning of Code LLMs on domain-specific service robot applications (due to their cost-effectiveness, better privacy protection, and customizability). However, open-weight LLMs are still prone to errors. As a result, using SELF-INSTRUCT naively can produce low-quality data, such as programs that violate domain-specific constraints or instructions that are infeasible for the robot to execute. For instance, as

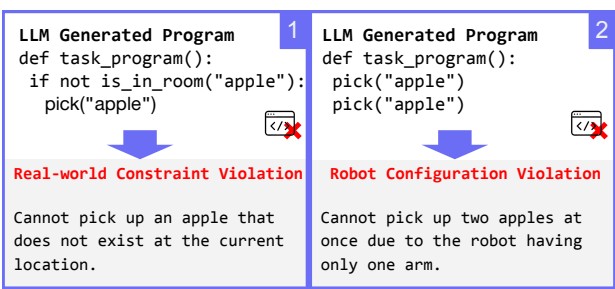

Figure 2: Examples of programs violating domain-specific constraints.

shown in the figure Fig. 2, the program may attempt to pick up an apple that is not present in the current location (example 1); or the instruction might ask the robot to pick up multiple objects simultaneously, which is physically impossible (example 2). To catch these errors, executing the program in a simulator with well-defined environments may appear promising. However, such environments require pre-enumeration of relevant entities and their states, which can result in rejecting a vast number of programs generated by SELF-INSTRUCT even if they are executable, simply because the pre-defined simulation environment fails to capture certain relevant aspects (*e.g.,* the presence of an "apple"); this undermines the diversity of the data generated.

This work introduces ROBO-INSTRUCT to bring the best of both worlds — it preserves the diversity of programs generated by an LLM while providing the correctness of simulator-based checking. ROBO-INSTRUCT is a new framework based on SELF-INSTRUCT, to address the challenges posed by using an open-weight LLM for generating domain-specific robot programs. As shown in Fig. 1, ROBO-INSTRUCT introduces two novel components: **(1)** ROBOSIM, a task-agnostic simulator that encodes domain-specific constraints and validates robot programs generated from SELF-INSTRUCT. Critically, ROBOSIM *dynamically synthesizes consistent simulation environments* starting from arbitrary programs. **(2)** INSTALIGN, an LLM-aided instruction-program alignment procedure that revises the generated instructions to better reflect the intent of the generated programs. ROBO-INSTRUCT also uses a rejection-sampling mechanism to discard invalid programs detected by ROBOSIM and query SELF-INSTRUCT for a new program based on the same instruction.

We use an 8B Llama3 model (AI, 2024) to generate instruction-program training data and fine-tune a 7B CodeLlama model (Rozière et al., 2024) on ROBOEVAL, a domain-specific benchmark for service robot programs. Our ROBO-INSTRUCT-fine-tuned model significantly outperforms the base model by 28.75% in pass@1 scores and achieves a 13.75% improvement over its SELF-INSTRUCT-fine-tuned variant. Moreover, it surpasses several larger code models, including Deepseek-Coder-33B (Guo et al., 2024), Starcoder2-15B (Lozhkov et al., 2024), GPT-3.5-Turbo (OpenAI, 2022), and Gemini-Pro (Team et al., 2024). Finally, in real-world deployment, we demonstrate that ROBO-INSTRUCT achieves significantly lower latency compared with GPT-3.5 and GPT-4.

**Contributions** Our main contributions are as follows

1. We introduce ROBO-INSTRUCT, a new framework for improving open-weight LLMs' ability to generate training data for finetuning Code LLMs on domain-specific service robot applications. This framework introduces two novel components, ROBOSIM and INSTALIGN.

2. We present ROBOSIM, a method to *synthesize consistent simulation environments dynamically* for verifying the generated programs. This method preserves the diversity of programs generated by an LLM while providing correctness with simulator-based checking.

3. We present INSTALIGN, a simple *instruction-program alignment procedure* that revises generated instructions better to reflect the actual results of the generated program.

4. We show that the ROBO-INSTRUCT finetuned model can significantly outperform the original base model and its finetuned variant using SELF-INSTRUCT in generating domain-specific service robot programs. It also surpasses several other larger Code LLMs.

## 2 ROBO-INSTRUCT

In this section, we describe how ROBO-INSTRUCT generates training datasets for domain-specific service robot programs. Fig. 1 provides a high-level overview of the framework. To add an entry to the training dataset, SELF-INSTRUCT first generates an instruction-program pair based on robot APIs and seed tasks (detailed prompts in Appendix A.4.2). Then, ROBOSIM dynamically synthesizes a consistent simulation environment to validate the program as it executes.

If the program is invalid, ROBO-INSTRUCT applies a rejection-sampling method, which generates a new program based on the same instruction and re-evaluates it. This process continues until the program is valid or a predefined maximum resampling limit is reached. If the limit is exceeded, the instruction may be incompatible with the domain-specific APIs or too complex, so the instruction-program pair is discarded.

Finally, if the program is valid, INSTALIGN uses an open-weight LLM to refine the instruction to better align with the program's intent, and the revised instruction-program pair is added to the training dataset. In the following subsections, we delve into the specific design of each component.

### 2.1 ROBOSIM: DYNAMIC SYNTHESIS OF SIMULATION ENVIRONMENTS FOR PROGRAM VALIDATION

We present a principled approach to designing ROBOSIM, a system for dynamically synthesizing consistent simulation environments to validate domain-specific robot programs. For service mobile robots, a simulation environment often relies on three concepts:

1. A list of **entities** to reason about, e.g., "apple", "kitchen"

2. The **type** of the entities, and hence their affordances, e.g., "apple" is an object, you can pick it up; "kitchen" is a location, you can go to it, and it contains objects.

3. The **state** of the entities in the world, e.g., the "apple" is in the "kitchen".

These concepts are closely related to the domain-specific APIs, where each API invocation during program execution can trigger updates to the simulation environment. To handle the interaction between APIs and simulation environments, we introduce DYNAMICEVAL, an algorithm inspired by Angelic Execution (Broy & Wirsing, 1981), a software engineering technique to infer program properties from incomplete API specifications. DYNAMICEVAL automatically generates a simulation environment for each program and checks its correctness within the inferred environment.

As shown in Alg. 1, upon each API invocation, the corresponding inputs and the current simulation environment are passed into DYNAMICEVAL. DYNAMICEVAL first infers relevant entities when they appear in the program being checked (line 5). For instance, if a program includes the statement pick("apple"), DYNAMICEVAL infers that apple is an entity to consider, even if it is not currently defined in the simulation environment.

For each relevant entity, if it has already been initialized (line 7), DYNAMICEVAL infers its type and new state from the API invocation and checks for any inconsistencies with the current simulation

---

**Algorithm 1** DYNAMICEVAL(api_fn, api_inputs, $\mathcal{W}$)

---

1: **Input:** api_fn                                          ▷ The API function name
2: **Input:** api_inputs                            ▷ The input received by the API invocation
3: **Input:** $\mathcal{W}$                                  ▷ The current simulation environment
4: info ← EXTRACT_API_INVOCATION_INFO(api_fn, api_inputs)
5: entities ← INFER_RELEVANT_ENTITIES(info)
6: **for** entity ∈ entities **do**
7:     **if** IS_ENTITY_INITIALIZED(entity, $\mathcal{W}$) **then**
8:         entity_type ← DEDUCE_TYPE(info)
9:         entity_new_state ← DEDUCE_STATE(info)
10:         **if** CHECK_TYPE_CONSISTENCY(entity, entity_type, info, $\mathcal{W}$) and \
            CHECK_STATE_CONSISTENCY(entity, entity_new_state, info, $\mathcal{W}$) **then**
11:            UPDATE_ENTITY_STATE(entity, info, $\mathcal{W}$)
12:         **else**
13:            **raise** "Error: state inconsistent or type mismatch"
14:         **end if**
15:     **else**
16:         INITIALIZE_ENTITY_WITH_RANDOM_STATE(entity, info, $\mathcal{W}$)
17:     **end if**
18: **end for**
19: retval ← GET_RETURN_VALUE(info, $\mathcal{W}$)
20: **return** retval

---

environment (lines 8-10). For example, pick requires an object type, while go_to requires a location type [1]. If a program contains:

```python
def task_program():
    pick("apple")
    go_to("apple")
```

By invoking API calls sequentially, DYNAMICEVAL first infers that "apple" is an object and then raises an error when go_to("apple") is called. If no inconsistency is detected, the simulation environment is updated accordingly based on the API definition (line 11).

On the other hand, if the entity has not been initialized, it will be assigned a random plausible state (line 16). For example, the API is_in_room(object) checks if an object is in the same location as the robot and expects a boolean return. In this case, DYNAMICEVAL assigns a 50% probability for the apple to be in the robot's current location — determining the state of the apple as either present or absent in the robot's location.

Finally, DYNAMICEVAL computes and returns the value based on the API specification and the updated simulation environment. Fig. 3 illustrates how ROBOSIM leverages DYNAMICEVAL to dynamically synthesize the simulation environment and validate the program. In ROBOSIM, the simulation starts with only the robot at its initial position, and entities are added as the program runs. At execution timestep 2a, after the robot picks up an apple, it becomes unclear whether another apple remains at the location. Therefore, DYNAMICEVAL sets the apple's state to "Undefined" and removes it in subsequent executions. This method is also related to STRIPS planning, as we demonstrate the connection in Appendix A.2.

In the example program shown in Fig. 3, it's clear to humans that the program's logic is flawed because it attempts to pick up the apple, which is not present in the room. But ***how would the simulator identify this as a failing state?*** ROBOSIM solves this issue by simulating all possible states of the discovered entities and verifying that none lead to erroneous program execution. In this case, the "apple" can either be present in the room or not. If the apple is not present, executing pick("apple") will result in an error. Checking all possible states requires exploring an exponentially growing number of combinations based on the entities discovered. To manage this, ROBOSIM

---

[1] In this example, type compatibility check is strict (i.e., "apple" is only an object and no further inference is made about its location). Nevertheless, the algorithm is also capable of handling more advanced scenarios.

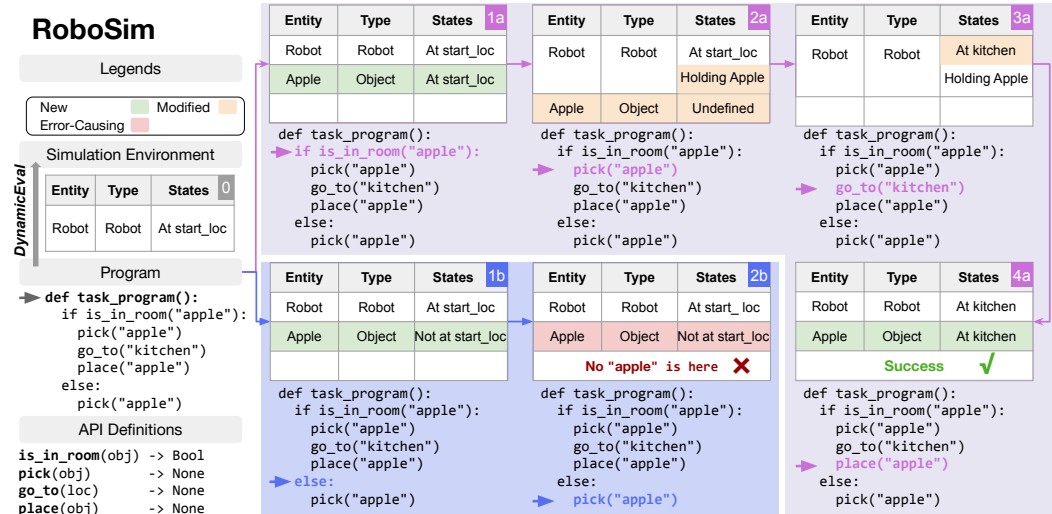

Figure 3: Illustration of ROBOSIM executing a task program and dynamically updating the simulation environment. Initially, the environment only contains the robot's starting location. As the program runs, two possible execution paths emerge (purple and blue). The environment is updated by DYNAMICEVAL at each step, reflecting the robot's actions, object states, and location changes along both paths.

employs a bounded compute budget to randomly sample from this exponential space, as detailed in the overall ROBOSIM algorithm in Alg. 2.

---

**Algorithm 2** ROBOSIM($\mathcal{P}$)

**Require:** Program $\mathcal{P}$                                                        ▷ Generated program
1: **Initialize:** Set $\mathcal{A}$                                          ▷ A set of domain-specific robot APIs
2: **Initialize:** $k$                                                         ▷ Number of evaluation iterations
3: **Initialize:** $\mathcal{W}_{\text{init}}$          ▷ An initial simulation environment with or without predefined information
4: $\mathcal{P}_{\text{trans}} \leftarrow$ TRANSLATE($\mathcal{P}, \mathcal{A}$, DYNAMICEVAL)          ▷ Replace each API call with DYNAMICEVAL
5: **for** $i = 1$ **to** $k$ **do**                                  ▷ Then, evaluate $\mathcal{P}$ $k$ times to catch program errors
6:     **try:**
7:         $\mathcal{W} \leftarrow \mathcal{W}_{\text{init}}$                              ▷ Initialize a new simulation environment
8:         $\text{exec}(\mathcal{P}_{\text{trans}}, \mathcal{W})$
9:     **catch:**
10:         **return** False
11: **end for**
12: **return** True                                              ▷ Return True if all program executions are successful

---

### 2.2 INSTALIGN: LLM-AIDED INSTRUCTION-PROGRAM ALIGNMENT PROCEDURE

Even after ROBOSIM verifies that a program doesn't violate domain-specific constraints, it may still have subtle inconsistencies with the instructions — such as omitting a step implied by the instruction. The proposed rejection-sampling strategy alone cannot resolve this issue, and correcting verified programs to fully reflect the instructions while ensuring they remain valid is a challenging task.

We note that LLMs nowadays have demonstrated impressive code *understanding* capabilities (Rozière et al., 2024; Nam et al., 2024; Leinonen et al., 2023; Li et al., 2023; Lekshmi-Narayanan et al., 2024). Instead of correcting the program, it may be more effective for the LLM to revise its generated instructions to better align with the program's intent.

We propose INSTALIGN, a procedure that prompts an LLM to revise generated instructions to better match the intent of the program, as shown in Fig. 4. INSTALIGN follows two steps: first, it uses Chain-of-Thought reasoning (CoT) (Wei et al., 2022) to generate a revised instruction based on the instruction-program pair; then, it prompts the LLM to compare the original and revised instructions,

Figure 4: Overview of INSTALIGN.

selecting the one that best aligns with the program's intent. Detailed prompts are provided in Appendix A.4.3.

# 3 ANALYSIS AND EXPERIMENTS

In this section, we investigate the following two research questions:

1. Is ROBO-INSTRUCT effective at generating training data to fine-tune a small language model to generate domain-specific programs for robots?

2. How do ROBOSIM and InstAlign impact the effectiveness of ROBO-INSTRUCT?

We conduct our investigation by fine-tuning the CodeLlama-Python-7B model (Rozière et al., 2024) on the synthetic dataset generated by ROBO-INSTRUCT and evaluate the fine-tuned model using ROBOEVAL (Hu et al., 2024), a domain-specific code generation benchmark for service mobile robots. In the following subsections, we first provide a brief description of ROBOEVAL. Then we present our experimental results addressing the two main research questions. Finally, we offer more analysis of ROBOSIM, INSTALIGN, and the synthetic dataset.

## 3.1 ROBOEVAL: A DOMAIN-SPECIFIC ROBOT CODE GENERATION BENCHMARK

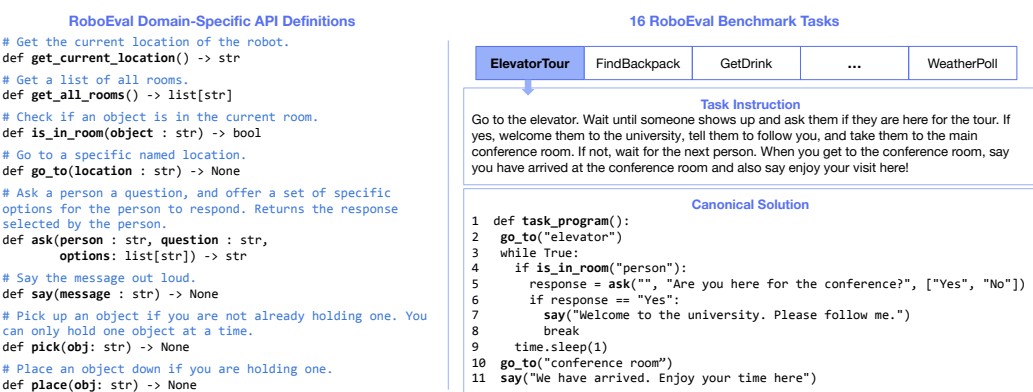

Figure 5: ROBOEVAL APIs and benchmark task example.

ROBOEVAL is a domain-specific code generation benchmark, featuring a suite of 16 tasks designed to evaluate the ability of LLMs to understand custom APIs and generate programs for service robots. In this domain, a service robot can perceive objects, navigate to various locations, manipulate items, and communicate with humans. Furthermore, the robot should be capable of basic commonsense reasoning and executing complex tasks that involve conditional and repetitive actions. To facilitate these capabilities, ROBOEVAL defines a set of 8 API functions in Python as skill primitives. Fig. 5

illustrates these function signatures and definitions, alongside an example task instruction and its canonical solution from the benchmark. In addition, unlike other popular code generation benchmark tasks (Chen et al., 2021; Austin et al., 2021; Li et al., 2022; Liu et al., 2023b; Lai et al., 2022; Hendrycks et al., 2021), **_the order of the robot's actions is crucial for successfully completing the specified tasks_**. For instance, in the task *"bring me a marker from the classroom that does not have a whiteboard,"* the robot must check each classroom until it finds one without a whiteboard, whereas simply bringing back a marker is insufficient. Hence, RoboEval evaluates the generated program by executing it in a simulator to capture the action traces, which are subsequently validated for sequence correctness using temporal logic.

## 3.2 RQ1: Is Robo-Instruct Effective at Generating Training Data to Fine-Tune a Small Language Model for Generating Domain-Specific Robot Programs?

| Fine-tune | Model | # Param | RoboEval pass@1 | | Licensing |
|---|---|---|---|---|---|
| | | | $T = 0$ | $T = 0.2$ | |
| - | GPT-4 | - | 83.75% | 85.81% | Proprietary |
| - | GPT-3.5-Turbo | - | 67.5% | 65.56% | Proprietary |
| - | Gemini-Pro | - | 60.00% | 59.88% | Proprietary |
| - | Codellama-Python | 34B | 46.25% | 48.25% | Open |
| - | Starcoder2 | 15B | 62.5% | 60.94% | Open |
| - | Deepseek-Coder | 33B | 53.75% | 52.13% | Open |
| - | Codellama-Python | 7B | 40.00% | 39.31% | Open |
| Self-Instruct | CodeLlama-Python | 7B | 55.00% | 52.69% | Open |
| Robo-Instruct (ours) | CodeLlama-Python | 7B | **68.75%** | **66.00%** | Open |
| - | Llama3 | 8B | 42.5% | 36.69% | Open |
| Self-Instruct | Llama-3 | 8B | 55.00% | 53.75% | Open |
| Evol-Instruct | Llama-3 | 8B | 57.5% | 54.87% | Open |
| Robo-Instruct (ours) | Llama-3 | 8B | **66.25%** | **62.44%** | Open |
| EI + RI (ours) | Llama-3 | 8B | **70.00%** | **66.38%** | Open |

Table 1: Pass@1 results of different LLMs on RoboEval computed with greedy decoding $T = 0$ and nucleus sampling $T = 0.2$.

**Experiment Setup.** We use the open-weight LLM, Llama3-8B-Inst, for Robo-Instruct. To generate a diverse dataset, we employ nucleus sampling for creating instruction-program pairs, setting the temperature $T = 1$ and top $p = 0.95$. The maximum resampling limit is capped at 3 to accommodate instructions that initially produce invalid programs. For the LLM used in InstAlign, we empirically adjust the generation temperature to $T = 0.3$ to optimize performance. Furthermore, we assess the edit similarity between token sequences of each instruction pair in the dataset (Lee et al., 2022), removing duplicates where the similarity score exceeds 0.6. The same similarity-based approach is used to decontaminate the dataset against the RoboEval benchmark tasks. We use the same setup to generate data via Self-Instruct. Instead of discarding invalid programs, Self-Instruct includes every generated instruction-program pair in the training dataset. Finally, we create two datasets with 5K instruction-program pairs each using Self-Instruct and Robo-Instruct respectively. These datasets are then used to fine-tune the CodeLlama-Python-7B model. The learning rate is set to be $3e$-5 with a warmup ratio of $3\%$ and a constant lr scheduler. We employ the AdamW optimizer (Loshchilov & Hutter, 2019) with an effective batch size of 8, training each model for 5 epochs using a sequence length of 2048 tokens. We train all our models on a single H-100 GPU using unsloth (Unslothai, 2024).

**Baselines.** We divide our baseline models into 2 categories: 1) proprietary LLMs, including GPT4 (OpenAI et al., 2024), GPT3.5-Turbo (OpenAI, 2022), Gemini-Pro (Team et al., 2024), and 2) open-weight LLMs, including Codellama-Python-7B (Rozière et al., 2024), Codellama-Python-34B, Starcoder2-33B (Lozhkov et al., 2024), Deepseek-Coder-33B (Guo et al., 2024), and Llama3-8B-Inst (AI, 2024).

Tab. 1 presents the average pass@1 results for different LLMs on RoboEval, using two settings: greedy decoding at temperature $T = 0$ and nucleus sampling at temperature $T = 0.2$. The results

show that ROBO-INSTRUCT-fine-tuned CodeLlama significantly improves upon the base CodeLlama-Python-7B, and outperforms the SELF-INSTRUCT-fine-tuned variant (Appendix A.3.3 shows that improvements over SELF-INSTRUCT are not the result of distributional biases in the selection process). Notably, ROBO-INSTRUCT surpasses all open-weight models, including larger ones like CodeLlama-Python-34B and Deepseek-Coder-33B. Additionally, although the training dataset was generated using Llama3-8B-Inst, which scores less than 50% pass@1 on ROBOEVAL from Tab. 1, our ROBO-INSTRUCT-fine-tuned model still achieves a significant improvement, scoring 68.75%. Finally, compared to proprietary models, while our ROBO-INSTRUCT-fine-tuned model trails the more powerful GPT-4, it outperforms GPT-3.5-Turbo and Gemini-Pro in generating programs for service mobile robots. This result demonstrates the effectiveness of our approach in generating domain-specific robot program data for fine-tuning a much smaller language model. It suggests that the fine-tuned model could potentially replace some proprietary models, providing a more cost-effective and private option for local deployment.

### 3.3 RQ2: How Do RoboSim and InstAlign Impact the Effectiveness of Robo-Instruct?

| Method | T=0 | | T=0.2 | | Invalid Programs |
|---|---|---|---|---|---|
| | pass@1 | Improv. | pass@1 | Improv. | |
| Codellama-7B-Python | 40.00% | +0% | 39.31% | +0% | 38.31% |
| SELF-INSTRUCT | 55.00% | +15.00% | 52.69% | +13.38% | 20.94% |
| +Reject Unsolvable (RU) | 60.00% | +20.00% | 57.62% | +18.31% | 23.38% |
| +ROBOSIM + RU | 63.75% | +23.75% | 63.88% | +24.57% | **14.13%** |
| +INSTALIGN + RU | 58.75% | +18.75% | 59.81% | +20.50% | 23.44% |
| +Both (ROBO-INSTRUCT) | **68.75%** | **+28.75%** | **66.00%** | **+26.69%** | 17.07% |

Table 2: Pass@1 results of different methods on ROBOEVAL computed with greedy decoding $T = 0$ and nucleus sampling $T = 0.2$. The **Invalid Programs** column indicates the percentage of programs that result in execution errors when tested on ROBOEVAL tasks.

Using the same setup as in the previous section, we investigate the effectiveness of ROBOSIM and INSTALIGN. Since SELF-INSTRUCT may generate instructions for which no corresponding valid program can pass in ROBOSIM, we include Reject Unsolvable (RU) as an additional baseline. SELF-INSTRUCT+RU discards instructions for which no valid programs were found to successfully execute in ROBOSIM, and preserves instructions for which at least one passing program was found. Tab. 2 shows the average pass@1 results from CodeLlama-7B-Python fine-tuned on different datasets generated by each method. First, results from SELF-INSTRUCT + RU indicate that simply discarding invalid instructions improves model performance. Additionally, fine-tuning with a dataset created from SELF-INSTRUCT + ROBOSIM results in the smallest proportion of invalid program errors. Finally, incorporating either ROBOSIM or INSTALIGN individually offers improvements over the baseline SELF-INSTRUCT + RU results, incorporating both in ROBO-INSTRUCT results in the best pass@1 performance. We refer the readers to Appendix A.1 for more results.

### 3.4 Qualitative analysis of the generated program errors

We qualitatively analyze invalid programs identified by ROBOSIM, as shown in Fig. 6. The first three examples are easily recognizable to humans as flawed. However, the last example is more complex and involves an error when the robot can navigate to more than two rooms. After the robot places a toy in the living room, DYNAMICEVAL updates the environment to reflect that a toy is now in the room (line 8). However, when the robot returns to the living room later (line 6), it will not drop the item it's holding (line 8). As a result, when the robot enters a third room (line 4) and tries to pick up another toy (line 5), an error will occur because the robot is only capable of carrying one item at a time. This example demonstrates that ROBOSIM can detect invalid programs beyond those easily identifiable through human inspection.

---

[2]Programs have been adapted to succinctly demonstrate the types of errors and fit within the figure.

```
LLM Generated Program                    [1]
1 def task_program():
2   go_to("game room")
3   if is_in_room("Jack"):
4     say("Hello Jack")
5   response = ask("Jack",
              "Want to play a game?",
              ["Yes", "No"])
```

**Real-world Constraint Violation**
Line 3 checks if Jack is in the room. If he is absent, line 5 raises an error, as it is illogical to ask Jack a question when he is not present.

```
LLM Generated Program                    [2]
1 def task_program():
2   list_of_rooms = get_all_rooms()
3   rooms_with_robots = []
4   for room in list_of_rooms:
5     if "robot" in is_in_room("robot"):
6       go_to(room)
```

**Return Value Violation**
Line 5 `is_in_room()` returns a boolean, which leads to a Python runtime error due to a type mismatch.

```
LLM Generated Program                    [3]
1 def task_program():
2   go_to("item storage room")
3   pick("item storage room")
```

**Entity Type Violation**
The robot cannot pick up a location.

```
LLM Generated Program                    [4]
1 def task_program():
2   for room in get_all_rooms():
3     if room != "living room":
4       go_to(room)
5       pick("toy")
6       go_to("living room")
7       if not is_in_room("toy"):
8         place("toy")
```

**Robot Configuration Violation**
If there are more than two rooms, the robot will attempt to pick up two toys, resulting in a violation of its configuration, as the robot is equipped with only one arm.

Figure 6: SELF-INSTRUCT-Generated Program Errors. Examples highlight errors that violate domain-specific constraints.[2]

# 4 REAL-WORLD DEPLOYMENT RESULTS

We deployed the ROBO-INSTRUCT fine-tuned model to generate and execute mobile robot programs in the real world, as shown in Fig. 7. Compared to GPT-4 and GPT-3.5-turbo (Tab. 3), our model generates programs about 6x faster than GPT-4 and 3x faster than GPT-3.5-turbo, with similar output quality. In Appendix A.5 we showcase more results on long-horizon tasks beyond ROBOEVAL.

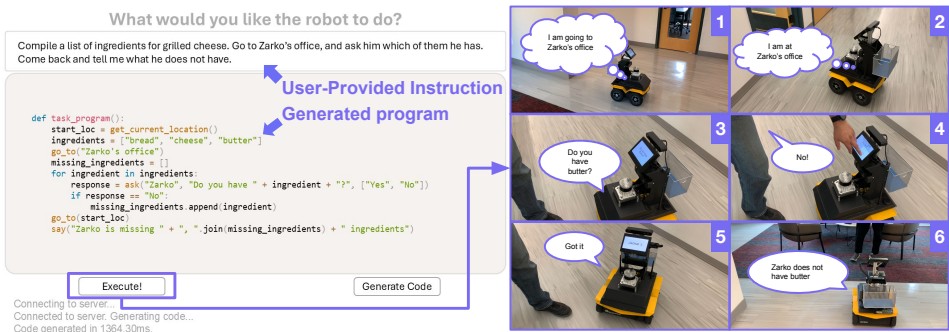

Figure 7: Deployment of the ROBO-INSTRUCT fine-tuned model to generate programs based on user-provided instructions and execute them on the robot.

| Models | GPT-4 | GPT-3.5-Turbo | Robo-Instruct (Ours) |
|---|---|---|---|
| **Inference Speed** | 19 tokens/s | 42 tokens/s | **114 tokens/s** |

Table 3: Inference speed of different models.

## 5 RELATED WORK

### 5.1 LLMS FOR ROBOT CODE GENERATION

LLMs have shown impressive capabilities in generating robot programs from natural language (Liang et al., 2022; Singh et al., 2023; Huang et al., 2023a). One popular approach uses LLMs to generate composable costmaps for robots to plan their motion on. In this approach, Voxposer (Huang et al., 2023b) focuses on the tabletop manipulation setting and NavCon (Biggie et al., 2023) focuses on creating composable maps for navigation. Using LLM to create reward functions is also promising. Eureka (Ma et al., 2023; 2024) and Language to Rewards for Robotic Skill Synthesis (Yu et al., 2023) both show that LLM can generate good reward functions that allows robots to acquire complex skills. Finally, LLM can also be used to generate programs for high-level planning. LLM+p (Liu et al., 2023a) outputs a robot plan in the form of the well-defined planning domain definition language (PDDL). Tidybot (Wu et al., 2023) uses an LLM to generate a rule that captures user preferences from examples and executes a program to sequentially complete the task in order. RoboEval (Hu et al., 2024) focuses on generating domain-specific programs for service mobile robots. It generates a program that allows the service robot to carry out long-horizon tasks and then validates the correctness of the program.

### 5.2 GENERATING DATASETS FOR FINE-TUNING LLMS

To enhance LLMs' performance in code generation, numerous studies have explored the creation of specialized datasets (Muennighoff et al., 2024; Köpf et al., 2023; Muennighoff et al., 2022). SELF-INSTRUCT (Wang et al., 2022) is one popular method for generating synthetic datasets using an LLM. Following this methodology, Alpaca (Taori et al., 2023) generates 52K instruction-following demonstrations and subsequently fine-tunes the LLaMA 7B model (Touvron et al., 2023) to create Alpaca 7B, which can behave qualitatively similarly to OpenAI's text-davinci-003. Code Alpaca (Chaudhary, 2023) extends this approach to generate code instructions using 21 seed tasks, while Gorilla-LM (Patil et al., 2023) adapts the method to focus on ML domain-specific APIs from Huggingface, TensorFlow Hub, and Torch Hub. To create more complex instructions, Evol-Instruct (Xu et al., 2024; Luo et al., 2024) proposes iteratively updating instructions to become more complex through different prompting strategies. In addition to Evol-Instruct, OSS-Instruct (Wei et al., 2023) uses open-source code snippets to generate 75K high-quality instruction data and fine-tunes the CodeLlama-Python-7B model to create Magicoder, which can match the performance of GPT-3.5-Turbo (OpenAI, 2022) on HumanEval (Chen et al., 2021). While these works focus on creating seed instruction sets to generate synthetic data to effectively fine-tune an LLM, our research investigates post-processing methods in addition to SELF-INSTRUCT. Specifically, we concentrate on generating domain-specific programs in robotics (Hu et al., 2024), where we can effectively leverage constraints to filter out erroneous programs.

## 6 CONCLUSION, LIMITATION AND FUTURE WORKS

In this work, we introduce ROBO-INSTRUCT, a novel framework to generate synthetic training data to fine-tune small language models for domain-specific robot programs. ROBO-INSTRUCT comprises two novel components: 1) ROBOSIM, a method to synthesize consistent simulation environments dynamically for verifying the generated programs, and 2) INSTALIGN, an LLM-aided instruction alignment procedure to revise instructions to better align with the generated programs. The experimental results show that the 7B CodeLlama model fine-tuned on the ROBO-INSTRUCT dataset significantly outperforms larger open-weight LLMs and proprietary models like GPT-3.5-Turbo and Gemini-Pro in generating service robot programs. However, a key limitation is that ROBO-INSTRUCT relies on SELF-INSTRUCT to filter invalid programs, which may introduce biases and affect dataset quality. Another limitation is the use of a simple rejection-sampling method to handle invalid programs, which may not fully address the underlying issues. Future work will focus on improving dataset quality by integrating ROBO-INSTRUCT with advanced methods like Evol-Inst and OSS-Inst.

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

# A APPENDIX

## A.1 OVERVIEW

In this appendix, we first outline the relationship between DYNAMICEVAL and the classic STRIPS planning formulation in subsection A.2, providing a new perspective on the proposed algorithm. In subsection A.3, we present additional ablation experiments to analyze the percentage of invalid programs generated by SELF-INSTRUCT and the effectiveness of the rejection-sampling strategy combined with ROBOSIM. We also explore how the generation temperature in INSTALIGN impacts final performance and compare the dataset diversity produced by ROBO-INSTRUCT and SELF-INSTRUCT. Subsection A.4 lists the seed tasks used in ROBOEVAL and the CoT prompts. in subsection A.5, we report real-world experiments that empirically evaluate the performance of our fine-tuned model on two long-horizon tasks, which differ significantly from those in ROBOEVAL, and assess the model's latency in generating programs. Although this work focuses on service mobile robots, the proposed framework is adaptable to other domains. In subsection A.6, we offer toy examples showing how DYNAMICEVAL can be extended to verify programs by incorporating domain-specific constraints.

## A.2 RELEVANCE TO STRIPS PLANNING

The proposed DYNAMICEVAL shares significant similarities with the formulation of STRIPS planning. A STRIPS instance is typically represented as a tuple $\langle I, G, A, P \rangle$, where $I$ denotes the initial state of the simulation environment, $G$ represents the desired goal state that the robot aims to achieve, $A$ defines the set of actions available to transition between states, and $P$ is the set of preconditions that must be satisfied before performing actions. Thus, DYNAMICEVAL can be reformulated to align with the STRIPS formulation as shown in Alg. 3. Each API invocation corresponds to an action, and its precondition consists of a set of literals, representing specific combinations of entities, types, and states.

To address this, we extend the classic STRIPS formulation by incorporating dynamically discovered literals. Unlike the conventional STRIPS approach, where each literal is binary—True when defined and False when not—we introduce a third value, "Undefined." This means a literal must be explicitly defined as either True or False; otherwise, it remains in the Undefined state. When an action requires a literal that is undefined, a random value (True or False) is assigned to it, and the literal is added to the state of the simulation environment (line 7). Once the precondition is fully defined, the action is executed, and domain-specific constraints are checked for any violations (line 10). This extension enables DYNAMICEVAL to handle arbitrary programs effectively.

---

**Algorithm 3** DYNAMICEVAL_STRIPS(api_fn, params, $\mathcal{W}$)

---

1: **Input:** api_fn               ▷ The API function name
2: **Input:** api_inputs          ▷ The input received by the API invocation
3: **Input:** $\mathcal{W}$          ▷ The current state of the simulation environment
4: $p \leftarrow$ GETPRECOND(api_fn, params)     ▷ Get the parameter-specific precondition for api_fn
5: **for** $l \in p$ **do**          ▷ Loop through every literal in the precondition
6:     **if** CHECKDEFINED($\mathcal{W}, l$) **is** Undefined **then**
7:        $\mathcal{W} \leftarrow$ GROWWORLD($l, \mathcal{W}$)     ▷ Randonly instantiate the literal and grow $\mathcal{W}$ to include it
8:     **end if**
9: **end for**
10: retval, $\mathcal{W} \leftarrow$ EXECUPDATE(api_fn, params, $\mathcal{W}$)     ▷ Execute api_fn and update $\mathcal{W}$
11: **return** retval

---

## A.3 ABLATION EXMPERIMENTS

### A.3.1 THE EFFECTIVENESS OF THE REJECTION-SAMPLING STRATEGY

We analyze the percentage of instruction-program pairs discarded by ROBOSIM at various maximum resampling limits, as shown in Fig. 8. Initially, with the maximum resampling limit set to 0, disabling the rejection-sampling method, approximately 51% of the programs generated by SELF-INSTRUCT contain errors. As the limit increases, fewer programs are discarded. However, there is a diminishing

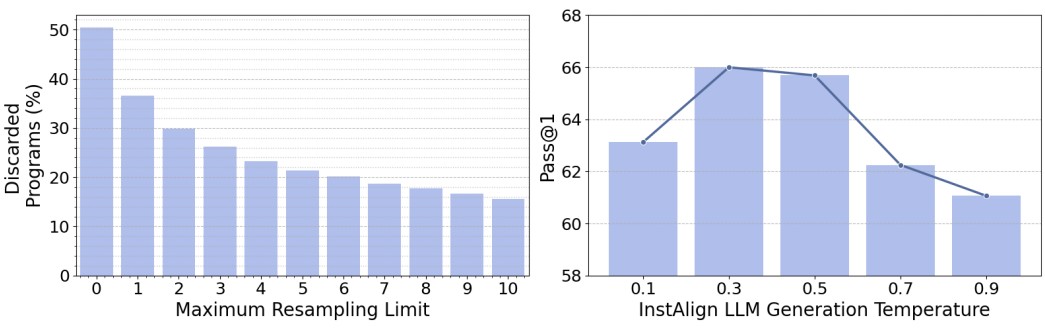

Figure 8: Ablation Experiment Results

return; even with the maximum resampling limit set to 10, about 15% of the instructions still result in invalid programs.

### A.3.2 INSTRUCTION ALIGNMENT MODEL TEMPERATURE

We further investigate how varying LLM temperatures for generating the revised instruction in INSTALIGN impact the performance of the fine-tuned model. Fig. 8 shows the bar chart of the pass@1 score of the models fine-tuned over datasets generated using different LLM temperatures in INSTALIGN. The model performs the best when fine-tuned on the dataset generated using LLM temperature $T = 0.3$. As the temperature increases, we observe a decrease in performance.

### A.3.3 ANALYSIS OF GENERATED DATASET

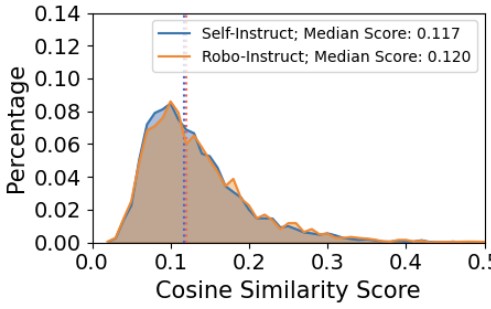

Figure 9: Cosine similarities between ROBOEVAL and generated data.

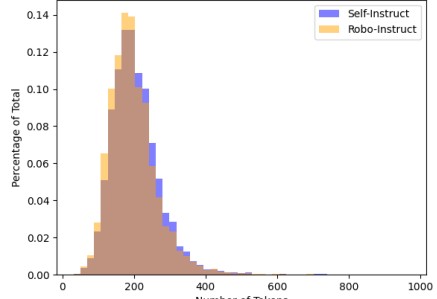

Figure 10: Token length distribution for each instruction-program pair.

Similar to Magicoder (Wei et al., 2023), we show the improvements from ROBO-INSTRUCT are not merely due to selection bias, i.e., including data more aligned with the distribution of ROBOEVAL tasks than SELF-INSTRUCT. We pair each sample from the generated dataset with task instructions and their canonical solutions, then compute cosine similarity using TF-IDF embeddings (Sparck Jones, 1988). Fig. 9 shows comparable token similarities between both methods. Additionally, Fig. 10 presents the token length distribution, which also appears similar for both.

| Method | Size | Ngram=4 Score | # Synth. Loc. | # Synth. Obj. |
|---|---|---|---|---|
| SELF-INSTRUCT | 5K | 0.581 | 956 | 1060 |
| ROBO-INSTRUCT | 5K | 0.587 | 1025 | 928 |

Table 4: Dataset Statistics

Since ROBOSIM does not rely on pre-defined simulation environments, we aim to assess the diversity of programs generated by SELF-INSTRUCT and whether ROBO-INSTRUCT can maintain this diversity.

To do so, we measure the number of distinct entities, such as synthetic locations and objects. As shown in Tab. 4, with a dataset of only 5,000 samples, approximately 1,000 unique objects and locations are generated, highlighting that conventional robot simulations with pre-defined environments are insufficient. Additionally, Tab. 4 presents the n-gram diversity scores for each dataset, indicating that both distributions and dataset statistics are highly similar. This suggests that ROBO-INSTRUCT not only preserves but enhances the quality of generated data compared to SELF-INSTRUCT, rather than simply aligning the dataset with benchmark tasks.

## A.4 PROMPTS

### A.4.1 ROBOEVAL SEED TASK EXAMPLE

Seed Task Example 1:

```python
# Instruction: Go to Arjun's office, ask him if he is ready to head out,
# and come back and tell me what he said
def task_program():
    start_loc = get_current_location()
    go_to("Arjun's office")
    response = ask("Arjun",
        "Are you ready to go?",
        ["Yes", "No"])
    go_to(start_loc)
    say("Arjun said: " + response)
```

Seed Task Example 2:

```python
# Instruction: Ask Alice if she needs 1, 2, or 3 boxes.
# Go to the storage room and ask if they have that many boxes.
# If so, go place the boxes in Alice's office.
# Otherwise, tell Alice you could not get the boxes.
def task_program():
    go_to("Alice's office")
    num_boxes = ask("Alice",
        "How many boxes do you need?",
        ["1", "2", "3"])
    go_to("storage room")
    response = ask("",
        "Do you have" + num_boxes + " boxes?",
        ["Yes", "No"])
    if response == "Yes":
        for _ in range(int(num_boxes)):
            pick("box")
            go_to("Alice's office")
            place("box")
            go_to("storage room")
    else:
        go_to("Alice's office")
        say("I could not get the boxes")
```

Seed Task Example 3:

```python
# Instruction: Check if there is a red marker in the main
# office, and if so, tell Eve that there is a marker there.
# If not, go to the supply room and
# bring a red marker to the main office.
def task_program():
    go_to("main office")
    red_marker_found = is_in_room("red marker")
```

```
8       if red_marker_found:
9           go_to("Eve's office")
10          say("There is a red marker in the main office")
11      else:
12          go_to("supply room")
13          pick("red marker")
14          go_to("main office")
15          place("red marker")
```

Seed Task Example 4:

```
1   # Instruction: Check every classroom if there is a whiteboard.
2   # Go to Aiden's office to tell him which room does not
3   # have a whiteboard. Come back and tell me task is completed.
4   def task_program():
5       start_loc = get_current_location()
6       list_of_rooms = get_all_rooms()
7       room_without_whiteboard = []
8       for room in list_of_rooms:
9           if "classroom" not in room:
10              continue
11          go_to(room)
12          if not is_in_room("whiteboard"):
13              room_without_whiteboard.append(room)
14      go_to("Aiden's office")
15      if len(room_without_whiteboard) > 0:
16          message = ""
17          for room in room_without_whiteboard:
18              message += room + ", "
19          message += "do not have a whiteboard"
20      else:
21          message = "all classrooms have a whiteboard"
22      say(message)
23      go_to(start_loc)
24      say("task is completed")
```

Seed Task Example 5:

```
1   # Instruction: Go to the kitchen and wait for someone
2   # to show up. When someone shows up, ask them to open
3   # the fridge, then pick up a diet coke.
4   # Finally, put the diet coke in the living room.
5   def task_program():
6       go_to("kitchen")
7       while True:
8           if is_in_room("person"):
9               response = ask("",
10                  "Please open the fridge",
11                  ["Yes", "No"])
12              if response == "Yes":
13                  pick("diet coke")
14                  break
15          time.sleep(1)
16      go_to("living room")
17      place("diet coke")
```

Seed Task Example 6:

```
1   # Instruction: Take a bed sheet from the laundry room
2   # and put it in each of the bedrooms.
```

```python
def task_program():
    start_loc = get_current_location()
    list_of_rooms = get_all_rooms()
    for room in list_of_rooms:
        if "bedroom" not in room:
            continue
        go_to("laundry room")
        pick("bed sheet")
        go_to(room)
        place("bed sheet")
    go_to(start_loc)
```

### A.4.2 PROMPTS TO GENERATE SYNTHETIC DATASET USING SELF-INSTRUCT

```
You are a helpful assistant. Here is a robot that has the
following capabilities:
- def get_current_location() -> str:
- def get_all_rooms() -> list[str]:
- def is_in_room(object : str) -> bool:
- def go_to(location : str) -> None:
- def ask(person : str, question : str, options: list[str]) -> str:
- def say(message : str) -> None:
- def pick(obj: str) -> None:
- def place(obj: str) -> None:
Generate an interesting robot task that can be accomplished using the
above capabilities.
{SEED EXAMPLE 1}

...

Generate an interesting robot task that can be accomplished using the
above capabilities.
{SEED EXAMPLE 6}

Generate an interesting robot task that can be accomplished using the
above capabilities.
```

### A.4.3 CoT PROMPTS FOR INSTALIGN

```
### Role
You are an expert at understanding robot programs.
You will be given a task instruction and robot program pair.
However, the instruction may not align with the program well.
You need to correct the task instruction to match the given robot program.

### Context
The robot only has access to the following 8 APIs and
standard Python functions
- def get_current_location() -> str:
- def get_all_rooms() -> list[str]:
- def is_in_room(object : str) -> bool:
- def go_to(location : str) -> None:
- ask(person : str, question : str, options: list[str]) -> str:
- say(message : str) -> None:
- def pick(obj: str) -> None:
- def place(obj: str) -> None:

### Inputs
Original Instruction
    This is a task instruction that may not align with the robot program
Robot Program
```

```
23      This is a python function starting with `def task_program():`
24
25  ### Task
26  1. Write down all the provided APIs used in the program and
    explain the effect of each API in this program
27
28  2. Examine these APIs and write down step by step what the program does
29  3. Combine all the results above and rewrite the instruction
30  You need to be specific and clear in your final corrected instruction.
```

## A.5 REAL-WORLD EXPERIMENTS

### A.5.1 PROGRAM GENERATION RESULTS BEYOND ROBOEVAL

We conducted a small qualitative experiment to assess the performance of the base model, SELF-INSTRUCT fine-tuned models, ROBO-INSTRUCT fine-tuned models, and GPT-4 on long-horizon tasks. To showcase performance, we created two new long-horizon tasks that differ significantly from those in the ROBOEVAL benchmark. Using a temperature setting of 0, we generated programs with each model and found that our ROBO-INSTRUCT fine-tuned model successfully produced correct programs, while both the base model and SELF-INSTRUCT fine-tuned model failed. Additionally, GPT-4 made an error on the second instruction. We show the generation results from ROBO-INSTRUCT fine-tuned models and GPT-4 below.

**Task instruction 1:**

```
1  Let's play a game: Double and give it to the next person. Start with
2  1 dollar. Go to rooms A, B, C, D, E, F, and G. If you see someone, tell
3  them how much money you have. Then ask if they would like to take the
4  money now or double the amount and give it to the next person. If they
5  choose to take it, the game is over, and you should come back to me.
6  Otherwise, double your money and continue. If, in the end, no one takes
7  the money, tell me how much you still have.
```

**ROBO-INSTRUCT fine-tuned model (Ours) [Success]**

```python
1  def task_program():
2      start_loc = get_current_location()
3      list_of_rooms = ["A", "B", "C", "D", "E", "F", "G"]
4      money = 1
5      for room in list_of_rooms:
6          go_to(room)
7          if is_in_room("person"):
8              say("I have $" + str(money))
9              response = ask("", """Would you like to take the money now
10             or double the amount and give it to the next person?""",
11             ["Take now", "Double and give"])
12             if response == "Take now":
13                 go_to(start_loc)
14                 say("The game is over, I gave the money to " + room)
15                 return
16             else:
17                 money *= 2
18     go_to(start_loc)
19     say("No one took the money, I still have $" + str(money))
```

**GPT-4 [Success]**

```python
1  def task_program():
2      start_loc = get_current_location()
```

```
3      rooms = ["A", "B", "C", "D", "E", "F", "G"]
4      money = 1
5      for room in rooms:
6          go_to(room)
7          if is_in_room("person"):
8              say("I have " + str(money) + " dollars")
9              response = ask("", """Would you like to take the money now
10                 or double the amount and give it to the next person?""",
11                 ["Take it", "Double it"])
12             if response == "Take it":
13                 say("Game over. The person in room " + room +
14                 " took the money.")
15                 go_to(start_loc)
16                 return
17             else:
18                 money *= 2
19     go_to(start_loc)
20     say("No one took the money. I still have " + str(money) + " dollars.")
```

## Task instruction 2:

```
1  Go to my office and check if I have a table, a chair, and a monitor
2  there. If any of these items are missing, go to Jason's office and
3  see if he is there. If he is, ask him if I can borrow the missing items.
4  If he agrees, pick up each missing item and bring it to my office.
5  If Jason is not in his office or he says no, come back and tell me the
6  reason.
```

## ROBO-INSTRUCT fine-tuned model (Ours) [Success]

```
1  def task_program():
2      start_loc = get_current_location()
3      go_to("my office")
4      table_found = is_in_room("table")
5      chair_found = is_in_room("chair")
6      monitor_found = is_in_room("monitor")
7      if not table_found:
8          go_to("Jason's office")
9          if is_in_room("Jason"):
10             response = ask("Jason", "Can I borrow the table from you?",
11             ["Yes", "No"])
12             if response == "Yes":
13                 go_to("Jason's office")
14                 pick("table")
15                 go_to("my office")
16                 place("table")
17             else:
18                 go_to(start_loc)
19                 say("Jason said no")
20         else:
21             go_to(start_loc)
22             say("Jason is not in his office")
23     if not chair_found:
24         go_to("Jason's office")
25         if is_in_room("Jason"):
26             response = ask("Jason", "Can I borrow the chair from you?",
27             ["Yes", "No"])
28             if response == "Yes":
29                 go_to("Jason's office")
30                 pick("chair")
31                 go_to("my office")
```

```
32              place("chair")
33          else:
34              go_to(start_loc)
35              say("Jason said no")
36      else:
37          go_to(start_loc)
38          say("Jason is not in his office")
39  if not monitor_found:
40      go_to("Jason's office")
41      if is_in_room("Jason"):
42          response = ask("Jason", "Can I borrow the monitor from you?",
43          ["Yes", "No"])
44          if response == "Yes":
45              go_to("Jason's office")
46              pick("monitor")
47              go_to("my office")
48              place("monitor")
49          else:
50              go_to(start_loc)
51              say("Jason said no")
52      else:
53          go_to(start_loc)
54          say("Jason is not in his office")
55  go_to(start_loc)
56  say("task is completed")
```

**GPT-4 [Failure]**

```
1  def task_program():
2      start_loc = get_current_location()
3      go_to("my office")
4      missing_items = []
5      for item in ["table", "chair", "monitor"]:
6          if not is_in_room(item):
7              missing_items.append(item)
8      if len(missing_items) > 0:
9          go_to("Jason's office")
10         if is_in_room("Jason"):
11             for item in missing_items:
12                 response = ask("Jason", "Can I borrow your " + item +
13                 "?", ["Yes", "No"])
14                 if response == "Yes":
15 # BUG: did not return to Jason's office after going back to my office
16                     pick(item)
17                     go_to("my office")
18                     place(item)
19                 else:
20                     go_to(start_loc)
21                     say("Jason did not allow me to borrow his " + item)
22                     return
23         else:
24             go_to(start_loc)
25             say("Jason is not in his office")
26             return
27     go_to(start_loc)
28     say("Task is completed")
```

## A.5.2 GENERATION LATENCY

**What would you like the robot to do?**

Go to the elevator. Wait until someone shows up and ask them if they are here for the tour. If yes, welcome them to the university, tell them to follow you, and take them to the main conference room. If not, wait for the next person. When you get to the conference room, say you have arrived at the conference room and also say enjoy your visit here!

```python
def task_program():
    start_loc = get_current_location()
    while True:
        go_to("elevator")
        if is_in_room("person"):
            response = ask("", "Are you here for the tour?", ["Yes", "No"])
            if response == "Yes":
                say("Welcome to the university! Please follow me.")
                go_to("main conference room")
                break
        time.sleep(1)
    say("I have arrived at the conference room. Enjoy your visit here!")
    go_to(start_loc)
```

| Execute! | Generate Code |
|---|---|

Connecting to server...
Connected to server. Generating code...
Code generated in 1162.00ms.        → Generated by our model

(a) Example Program 1 generated by our fine-tuned model.

**What would you like the robot to do?**

Go to the elevator. Wait until someone shows up and ask them if they are here for the tour. If yes, welcome them to the university, tell them to follow you, and take them to the main conference room. If not, wait for the next person. When you get to the conference room, say you have arrived at the conference room and also say enjoy your visit here!

```python
def task_program():
    go_to("elevator")
    while True:
        if is_in_room("person"):
            response = ask("", "Are you here for the tour?", ["Yes", "No"])
            if response == "Yes":
                say("Welcome to the university! Please follow me to the main conference room.")
                go_to("main conference room")
                say("We have arrived at the main conference room. Enjoy your visit here!")
                break
        time.sleep(1)
```

| Execute! | Generate Code |
|---|---|

Connecting to server...
Connected to server. Generating code...
Code generated in 6822.60ms.        → Generated by GPT-4

(b) Example Program 1 generated by GPT-4.

Figure 11: Our fine-tuned model is approximately 6x faster in inference speed than GPT-4 (Part 1).

(c) Another Example Program generated by GPT-4.

(d) Another Example Program generated by GPT-4.

Figure 12: Our fine-tuned model is approximately 6x faster in inference speed than GPT-4 (Part 2).

## A.6 TOY EXAMPLES BEYOND SERVICE MOBILE ROBOTS

### A.6.1 ROBOT WITH LOW-LEVEL CONTROLS

Consider a tabletop manipulation scenario with a potential API function, is_rotate(robot_gripper_name, radians), where the robot's gripper has a physical constraint, allowing rotation only within the range $\left[-\frac{\pi}{6}, \frac{\pi}{6}\right]$ radians. For the following generated program snippet:

```python
def task_program():
    rotate("left hand", math.pi/6)
    rotate("left hand", math.pi/6)
    rotate("left hand", math.pi/6)
```

DYNAMICEVAL will first infer that "left hand" is an entity of the robot gripper type. Then, regardless of the initial configuration of the gripper, DYNAMICEVAL will throw an error because the program causes the gripper to exceed its allowable range of motion.

### A.6.2 AI-POWERED PERSONAL DIGITAL ASSISTANT

Consider a broader application than robotics: code generation for an AI-powered personal digital assistant. This AI assistant could handle scheduling events using an API function like schedule_on_calendar(event, start_time, duration). Given the instruction: *"My schedule is free tomorrow morning. Please create two 1-hour timeslots for office hours for my robotics and deep learning class."* The assistant could generate a program to create these timeslots:

```python
def task_program():
    schedule_on_calendar("robotics class office hour",
                         "9:30 am", "1 hr")
    schedule_on_calendar("deep learning class office hour",
                         "10:00 am", "1 hr")
```

In this example, DYNAMICEVAL needs to reason about the entities "robotics class office hour" and "deep learning class office hour", which are categorized as event types. The event type indicates that these entities have associated timeslots. The state of these entities is defined by the time they occur: robotics class office hour is set for 9:30-10:30 am, and deep learning class office hour is set for 10:00-11:00 am. During evaluation, DYNAMICEVAL can identify a time conflict between these two office hours and thus determine that the generated program is invalid.

