# OpenReview forum: "ROBO-INSTRUCT: Simulator-Augmented Instruction Alignment For Finetuning Code LLMs"
_ICLR.cc/2025/Conference — Submitted to ICLR 2025_

### Official Review · Reviewer_dWpD · 2024-10-18

**Soundness:** 1
**Presentation:** 2
**Contribution:** 2
**Rating:** 3
**Confidence:** 3

**Summary:**

This paper proposes a framework (Robo-Instruct) to generate training data to fine-tune a code LLM for domain-specific service robot applications. Robo-Instruct contains 2 components: (1) RoboSim that dynamically synthesizes consistent simulation environments for each generated program, and (2) InstAlign that handles the instruction-program inconsistencies. In the experiments, the authors use Llama3-8B-Instruct as the LLM to generate training data, and fine-tuned CodeLlama to perform on the RoboEval benchmark.

I think the authors are tackling an important problem that would allow LLMs to be better applied to robotics. However, I find the paper hard to follow, with insufficient experiments to support the potentially over-claimed contributions. Please see my explanation below.

**Strengths:**

- Having a simulation (not a physics engine in this paper) to improve the diversity of the generated programs intuitively increases the performance of an LLM for robotic applications, the paper is therefore addressing problems of importance.

**Weaknesses:**

- The contributions of the paper are limited. Of the 5 contributions listed in the introduction, I find 4 of them questionable. (1) RoboSim is said to produce diversity, yet on line 245 the authors explain that checking all possible states is not possible and they resort to random sampling with limited compute budget. This does not necessarily guarantee diversity. (2) InstAlign is just CoT. (3) The authors claim the fine-tuned model is better, but it is not clear how different are the data it has been trained on from the tasks it is tested in. (4) The authors claim the fine-tuned model is faster in inference than proprietary models. This is especially unfair and misleading. Any onboard model with sufficient hardware support is faster than remote API calls.
- Insufficient experiments. How different are the generated programs from the tasks in RoboEval? Is it a generalized performance or performance on the training set? Why use Llama3-8B-Instruct as the generating LLM and CodeLlama as the fine-tuned LLM? If you change them to other LLMs, will the framework still work?
- Presentation can be significantly improved. The paper is full of details that do not help a reader follow the main thread of the paper. None of the links (e.g., citations, sections, figures, etc) works, and I have to manually search for those entities. Inefficient use of space (e.g., are the 2 python programs on lines 188 and 202 necessary?).

**Questions:**

- How different are the generated programs from the tasks in RoboEval? Is it a generalized performance or performance on the training set?
- Why use Llama3-8B-Instruct as the generating LLM and CodeLlama as the fine-tuned LLM? If you change them to other LLMs, will the framework still work?

---

> ### Author Response · Authors · 2024-11-26
> **Response to Reviewer dWpD**
>
> We appreciate the reviewer's constructive feedback. In response, we have conducted additional experiments to include in the paper. Below are our responses to the points raised in the reviews:
>
>
> **Weakness:** The contributions of the paper are limited. Of the 5 contributions listed in the introduction, I find 4 of them questionable. (1) RoboSim is said to produce diversity, yet on line 245 the authors explain that checking all possible states is not possible and they resort to random sampling with limited compute budget. This does not necessarily guarantee diversity. (2) InstAlign is just CoT. (3) The authors claim the fine-tuned model is better, but it is not clear how different are the data it has been trained on from the tasks it is tested in. (4) The authors claim the fine-tuned model is faster in inference than proprietary models. This is especially unfair and misleading. Any onboard model with sufficient hardware support is faster than remote API calls.
>
> **Response**
> 1. The diversity refers to the variety of programs being generated. We measure diversity by counting the different numbers of agents, objects, and locations in the generated programs, as shown in Appendix A3.3 Table 4. On the other hand, state sampling aims to verify program correctness. Since programs can have different logical structures (e.g., using if statements), state sampling ensures that all parts of the program function correctly, rather than introducing diversity in the generated programs.
>
> 2. While InstAlign uses the Chain-of-Thought (CoT) approach, **using CoT naively does not work**: re-prompting the task instruction to generate a new program often results in invalid programs. This offsets the benefits of RoboSim and fails to produce correct outputs. We propose using CoT to rephrase instructions rather than generate new programs. This approach aligns instructions more effectively and takes advantage of modern LLMs, which are extensively trained in code understanding. Our key contribution is the novel way CoT is applied in our work
>
> 3. The reported results reflect performance on the test set, not the training set. Additionally, in Appendix A.5, we evaluate the model on two complex scenarios beyond RoboEval, demonstrating that the fine-tuned model performs well even in challenging scenarios.
>
> 4. The claim about inference speed is intended to highlight the practical advantages of finetuning models over relying on remote APIs. We will revise the text to ensure this point is framed as an explanation of the motivation behind this work, rather than a direct performance comparison.
>
>
> **Weakness:** Insufficient experiments. How different are the generated programs from the tasks in RoboEval? Is it a generalized performance or performance on the training set?
>
> **Response**
>
> We have applied filtering methods to deduplicate and decontaminate the generated dataset, as detailed in the Experiment Setup section and Appendix A3.3. The reported results reflect performance on the test set, not the training set. Additionally, in Appendix A.5, we evaluate the model on two complex scenarios beyond RoboEval, demonstrating that the fine-tuned model performs well even on these challenging tasks.
>
> **Weakness:** Presentation can be significantly improved. The paper is full of details that do not help a reader follow the main thread of the paper. None of the links (e.g., citations, sections, figures, etc) works, and I have to manually search for those entities. Inefficient use of space (e.g., are the 2 python programs on lines 188 and 202 necessary?).
>
> **Response**
>
> Thank you for the suggestions. We have updated the paper to reflect the changes.
>
> **Question:** Why use Llama3-8B-Instruct as the generating LLM and CodeLlama as the fine-tuned LLM? If you change them to other LLMs, will the framework still work?
>
> **Response**
>
> We chose to finetune CodeLlama due to its specialization in code generation. Additionally, we have updated the work with finetuning experiments on LLaMA3, as detailed in Table 1, which shows that Robo-Instruct consistently outperforms the Self-Instruct baseline.

---

> > ### Comment · Reviewer_dWpD · 2024-11-30
> >
> > Thank you for the responses, however, I still find the quality of the paper below the acceptance bar. For example:
> > 1. *"We propose using CoT to rephrase instructions rather than generate new programs."* This is a limited contribution in my opinion.
> > 2. *"Additionally, in Appendix A.5, we evaluate the model on two complex scenarios beyond RoboEval, demonstrating that the fine-tuned model performs well even in challenging scenarios."* Two is a very small number and can be easily cherry-picked.
> > 3. Representation still has significant room for improvement. To start with, table captions should be placed on top of the tables (see [guideline](https://arxiv.org/html/2410.02646v2)). In Algorithm 3, the func declaration accepts (api_fn, params, w), but the inputs on lines 1-3 are listed as api_fn, api_inputs, W. These issues suggest that the authors did not dedicate sufficient time to
> >  improve the paper's readability. I also wish the authors had made changes in a different color, it would reduce a reviewer's burden of reading and comparing.
> >
> > For these reasons, I'll keep my current rating.

---

### Official Review · Reviewer_nfRk · 2024-10-27

**Soundness:** 2
**Presentation:** 3
**Contribution:** 3
**Rating:** 5
**Confidence:** 4

**Summary:**

The paper introduces ROBO-INSTRUCT, a framework designed to enhance the generation of training data for fine-tuning Code LLMs in domain-specific service robot applications. It consists of ROBOSIM, a task-agnostic simulator that dynamically creates consistent simulation environments to verify program correctness, and INSTALIGN, an LLM-aided instruction-program alignment process. The framework significantly improves the performance of a fine-tuned model, achieving a 28.75% improvement in pass@1 over the base model and surpassing several proprietary models. Additionally, ROBO-INSTRUCT demonstrates faster inference speeds, making it suitable for real-world robot deployments.

**Strengths:**

1. Applying code to embodied AI is an important direction, so exploring how to enhance the code generation capabilities of LLMs in the robotics domain is also meaningful.
2. The idea of guiding the synthesized data with verification of the ROBOSIM environment is reasonable.
3. The experimental result looks promising.

**Weaknesses:**

I'd be happy to raise my score if my concerns are addressed.
1. There are many instruction tuning methods for code LLMs, but this paper only compares with SELF-INSTRUCT. Could the authors compare with methods like evol-instruct in Wizardcoder as well?
2. The choice of settings is somewhat confusing. For example, why use the Llama3 model to generate the training set and then fine-tune the 7B CodeLlama instead of Llama3 itself? Why not use more powerful closed-source models like GPT-3.5 or GPT-4-Turbo for synthesizing the dataset? I think the authors should either provide corresponding results or at least explain the reason for doing so.
3. I think the author should have decontaminated the test set, but it seems the author did not mention any relevant details in the experimental setup.
4. The real-world deployment results are great but the authors only measured the inference speed. Could the authors measure some accuracy or success-related metrics?

**Questions:**

1. I'm a bit confused about ROBOSIM. It is claimed that ROBOSIM is task-agnostic but it seems that ROBOSIM still requires a lot of expert knowledge for the corresponding task. For example, ROBOSIM cannot work on the "apple" tasks if the "apple" or "kitchen" related properties (i.e., missing any of the entities, type, or state) are absent. Or, to put it another way, even if "apple" and "kitchen" are present and ROBOSIM can complete tasks related to "apple" and "kitchen," it still won't be able to complete tasks related to "apple" and "living room" due to the absence of the "living room."
2. As the above question said, how much expert effort is needed to construct ROBOSIM?
3. A typo in line 375 "Gemino-Pro".
4. In Table 2, why does ROBO-INSTRUCT have a higher invalid program rate than ROBOSIM + RU?

---

> ### Author Response · Authors · 2024-11-26
> **Response to Reviewer nfRk**
>
> We appreciate the reviewer's constructive feedback. In response, we have conducted additional experiments to include in the paper. Below are our responses to the points raised in the reviews:
>
> **Weakness:** There are many instruction tuning methods for code LLMs, but this paper only compares with SELF-INSTRUCT. Could the authors compare with methods like evol-instruct in Wizardcoder as well?
>
> **Response**
>
> We have performed an additional experiment applying Evol-Instruct to fine-tune models for generating robot programs (the results have been updated in Table 1). It is clear that Evol-Instruct does not perform much better than Self-Instruct, whereas Robo-Instruct still provides a significant improvement for the task.
>
> **Weakness:** The choice of settings is somewhat confusing. For example, why use the Llama3 model to generate the training set and then fine-tune the 7B CodeLlama instead of Llama3 itself? Why not use more powerful closed-source models like GPT-3.5 or GPT-4-Turbo for synthesizing the dataset? I think the authors should either provide corresponding results or at least explain the reason for doing so.
>
> **Response**
>
> We have conducted finetuning experiments with LLaMA3, updated in Table 1. The results show that Robo-Instruct continues to outperform the Self-Instruct baseline. The focus on smaller open-weight models is due to their advantages in many applications, including speed, cost-effectiveness, customizability, and privacy preservation. Lastly, we believe that our methods show superior robustness when the programs are ***not*** synthesized by a much stronger model like GPT-4, but rather, are completely self-sufficient on open models.
>
>
> **Weakness:** I think the author should have decontaminated the test set, but it seems the author did not mention any relevant details in the experimental setup.
>
> **Response**
>
> Yes, please refer to the Experiment Setup section and Appendix A3.3, where we applied filtering methods to deduplicate and decontaminate the dataset. We have revised the description in these sections to make it more explicit.
>
>
> **Weakness:** The real-world deployment results are great but the authors only measured the inference speed. Could the authors measure some accuracy or success-related metrics?
>
> **Response**
>
> These test cases are part of RoboEval already. This work focuses on improving the code generation capabilities of large language models (LLMs) for programming service mobile robots. The skill APIs used in this study consist of high-level robot commands, and the correctness of the generated programs (assessed in RoboEval) determines their correctness via various checks.  This work does not look into low-level robot controllers.

---

> ### Author Response · Authors · 2024-11-26
> **Response to Reviewer nfRk (Part 2)**
>
> **Question:** I'm a bit confused about ROBOSIM. It is claimed that ROBOSIM is task-agnostic but it seems that ROBOSIM still requires a lot of expert knowledge for the corresponding task. For example, ROBOSIM cannot work on the "apple" tasks if the "apple" or "kitchen" related properties (i.e., missing any of the entities, type, or state) are absent. Or, to put it another way, even if "apple" and "kitchen" are present and ROBOSIM can complete tasks related to "apple" and "kitchen," it still won't be able to complete tasks related to "apple" and "living room" due to the absence of the "living room."
>
> **Response**
>
> When we describe ROBOSIM as task-agnostic, we mean that it can verify arbitrarily generated programs within the bounds of domain-specific constraints. These constraints, which require expert knowledge as discussed in the paper, refer to a common-sense understanding of the physical world. For example, "the robot cannot pick up an object that does not exist in the environment." Domain experts are responsible for manually encoding such constraints into ROBOSIM.
>
> Beyond these predefined constraints, no further information is required from the expert and ROBOSIM can dynamically infer other program-specific details during execution. For instance, as detailed in Algorithm 1, all entities are initially assumed to be unknown. When a program executes a line referencing specific information, such as a "living room," ROBOSIM checks the simulator's current state to verify if the entity is already defined (as either existent or non-existent). If the entity is undefined, ROBOSIM adds this information to the simulation environment. This dynamic updating capability ensures that ROBOSIM can verify arbitrarily generated programs.
>
> **Question:** As the above question said, how much expert effort is needed to construct ROBOSIM?
>
> **Response**
>
> In robotics applications, relevant constraints include a common-sense understanding of the physical world, such as "the robot cannot pick up an object that does not exist in the environment," and knowledge of the robot's configurations, such as "the robot has only one arm and can hold only one item at a time." Designing these constraints is not overly demanding. To illustrate further, we provide two toy examples in Appendix A.6 that demonstrate how constraints can be designed for other application domains.
>
> **Question:** A typo in line 375 "Gemino-Pro".
>
> **Response**
>
> Thank you for pointing it out. We have fixed this in the updated paper.
>
> **Question:** In Table 2, why does ROBO-INSTRUCT have a higher invalid program rate than ROBOSIM + RU?
>
> **Response**
>
> Thank you for pointing out this observation. While the exact reason is unclear, one possible explanation could be due to variance in the training process.

---

> > ### Comment · Reviewer_nfRk · 2024-12-02
> >
> > I thank the authors for their responses! Some of my concerns have been addressed. However, the remaining concerns mainly involve Table 1, expert effort, and the real-world deployment experiment.
> >
> > **About Table 1.** I noticed the experiments added by the authors. However, first, the details of the newly added experiments don't seem to be fully presented (e.g., evol-instruct). Second, I am somewhat confused by EI+RI achieving the best results, as this makes the contribution of this work unclear. Based on this result, are the authors suggesting that EI+RI is the final method of this work?
> >
> > **Expert overhead.** I have read Appendix A.6 but still couldn’t grasp how much expert effort is required to set these constraints. For instance, in the experiments presented in the paper, how much total expert effort is needed to establish these constraints?
> >
> > **Real-world robot experiments.** The authors mentioned that "these test cases are part of RoboEval already." Does this mean that tasks that can be completed in RoboEval can also be completed by real robots? Are these tasks unaffected by real-world conditions, such as the spatial layout of the kitchen? Could the authors conduct real-world task success rate tests, similar to inference speed tests, and provide convincing results?

---

### Official Review · Reviewer_d5Tg · 2024-11-04

**Soundness:** 3
**Presentation:** 3
**Contribution:** 2
**Rating:** 5
**Confidence:** 4

**Summary:**

The paper introduces ROBO-INSTRUCT, a framework designed to improve open-weight Large Language Models (LLMs) for generating domain-specific training data. This framework aims to enhance service robot applications, focusing on Code LLMs that use domain-specific APIs to generate executable robot instructions. The ROBO-INSTRUCT framework comprises two main components: ROBOSIM: A task-agnostic simulator to synthesize simulation environments for verifying program correctness dynamically. INSTALIGN: An alignment tool that adjusts program instructions to reflect the true intent of generated programs, using a large language model (LLM) for instruction revision. Experiments show that models fine-tuned with ROBO-INSTRUCT outperform base models and models fine-tuned with SELF-INSTRUCT, improving pass@1 scores and real-world deployment latency.

**Strengths:**

Clarity: The framework's purpose, components, and experimental results are presented clearly, though some complex aspects could benefit from additional clarification (e.g., the alignment between ROBOSIM and traditional STRIPS planning). The experiment design is well-articulated, showing comparisons across multiple baselines and careful control of variables.

Novelty: The integration of ROBOSIM for dynamic simulation and the INSTALIGN for instruction alignment introduce a novel approach to overcoming LLM limitations in handling domain-specific instructions. The work holds promise for cost-effective deployment of LLMs in real-world robot applications, especially where open-weight models are prioritized for privacy and customizability.

**Weaknesses:**

(1) The data augmentation approach seems somewhat incremental, given its widespread use in Evol-Instruct, WizardLM, and similar frameworks. It would be valuable to explore more unique challenges and solutions tailored to robotics, which often requires handling more complex tasks. Additionally, an evaluation on scaling performance regarding parameter count, generalization, and related metrics would strengthen the analysis.

(2) Another concern is that the evaluated tasks in the paper appear overly simplified. While I understand that many current studies also rely on simplified environments like VirtualHome, the solution's handling of out-of-distribution scenarios remains insufficiently understood. This is a crucial factor for robotics applications, where the risk of overfitting due to augmentation is particularly high.

**Questions:**

(1) How does ROBO-INSTRUCT handle edge cases where simulator-based validation cannot capture subtler domain inconsistencies?

(2) relying on data augmentation techniques such as self-instruct or evol-instruct may introduce bias or even hurt the generalization of LLMs, it would be nice to see related evaluation on

(3) the paper verifies the program correctness, is there any other filtering like ROGUE-L methods as used in self-instruct?

---

> ### Author Response · Authors · 2024-11-26
> **Response to Reviewer d5Tg**
>
> We appreciate the reviewer's constructive feedback. In response, we have conducted additional experiments to include in the paper. Below are our responses to the points raised in the reviews:
>
> **Weakness:** The data augmentation approach seems somewhat incremental, given its widespread use in Evol-Instruct, WizardLM, and similar frameworks. It would be valuable to explore more unique challenges and solutions tailored to robotics, which often requires handling more complex tasks. Additionally, an evaluation on scaling performance regarding parameter count, generalization, and related metrics would strengthen the analysis.
>
> **Response**
>
> We appreciate the reviewer’s insightful comments. While frameworks like Evol-Instruct have shown remarkable results in generating diverse data, our updated experiments reveal that Evol-Instruct alone is insufficient to enhance LLM performance. As shown in Table 1, Evol-Instruct performs only marginally better than Self-Instruct, whereas Robo-Instruct  can deliver significant improvements for the task. In addition, while exploring unique challenges and solutions specific to robotics and evaluating scaling performance across metrics such as parameter count and generalization are valuable directions, they fall outside the scope of this work. We hope to address these aspects in future research.
>
> **Weakness:**  Another concern is that the evaluated tasks in the paper appear overly simplified. While I understand that many current studies also rely on simplified environments like VirtualHome, the solution's handling of out-of-distribution scenarios remains insufficiently understood. This is a crucial factor for robotics applications, where the risk of overfitting due to augmentation is particularly high.
>
> **Response**
>
> We appreciate the reviewer’s concern. In Section A.5, we evaluate two additional complex scenarios beyond RoboEval. The fine-tuned model performs well on these challenging out-of-distribution tasks. This provides promising evidence of its robustness and ability to mitigate overfitting risks.
>
> **Question:** How does ROBO-INSTRUCT handle edge cases where simulator-based validation cannot capture subtler domain inconsistencies?
>
> **Response**
>
> The way Robo-Instruct verifies programs depends on the pre-specified domain-specific constraints defined by developers (we have provided two toy examples demonstrating how such constraints can be designed for other application domains in Appendix A.6). Robo-Instruct does not specifically address edge cases where simulator-based validation fails to capture subtler domain inconsistencies, and such cases may be included in the dataset. However, these edge cases are rare and are unlikely to have a significant impact on the final dataset.
>
>
> **Question:** Relying on data augmentation techniques such as self-instruct or evol-instruct may introduce bias or even hurt the generalization of LLMs, it would be nice to see related evaluation on
>
> **Response**
>
> We have performed an additional experiment applying Evol-Instruct to fine-tune models for generating robot programs (the results have been updated in Table 1). It is clear that Evol-Instruct does not perform much better than Self-Instruct, whereas Robo-Instruct still provides a significant improvement for the task.
>
> **Question:** the paper verifies the program correctness, is there any other filtering like ROGUE-L methods as used in self-instruct?
>
> **Response**
>
> Yes, please refer to the Experiment Setup section and Appendix A3.3, where we applied filtering methods to deduplicate and decontaminate the dataset. We have revised the description in these sections to make it more clear.

---

### Official Review · Reviewer_ycyR · 2024-11-04

**Soundness:** 4
**Presentation:** 4
**Contribution:** 2
**Rating:** 6
**Confidence:** 2

**Summary:**

The goal of this paper is to

**Strengths:**

Clarity. The authors did a phenomenal job describing their method and experimental process with precision. The specifics of the robosim environments were easy to follow from the method section and the motivation for dynamic environment generation and its unique application to robotic service agents was well presented.

Quality. The approach is simple and sound and I believe there is sufficient information for researchers to reproduce the results. On the RoboEval benchmark their method produces a model that outperforms even proprietary language models.

Significance. The idea of using dynamic environments to evaluate code could have broad impact for robotic code generation. The author present a strong first demonstration of this.

**Weaknesses:**

Presentation. Focusing solely on open models as being error-prone unnecessarily limits the scope and potential impact of the paper's contributions when it could be relevant to any base LLM.

Quality. There was a limited diversity of baselines. The domain specific language looks very similar to the code-as-policies test environments. In the experimental section, the authors should contextualize the results by either explaining the best analogy to code-as-policies that they run or by explicitly discussing a code-as-policies style baseline. The authors could also try simpler variants of their method: for example, taking some data points generated by robosim rejection sampling process and putting them in the prompt instead of doing model fine-tuning and alignment.

Significance. The APIs in the roboeval benchmark are very high-level and I wouldn't be surprised if there are many solid engineering approaches to getting a performant policy that works with high-level APIs. It's unclear how well roboinstruct will scale with more complex APIs and tasks, which limits the significance of the work.

**Questions:**

Could you please fix the formatting of table 5 to remove the overlapping text?

---

> ### Author Response · Authors · 2024-11-26
> **Response to ycyR**
>
> We appreciate the reviewer's constructive feedback. In response, we have conducted additional experiments to include in the paper. Below are our responses to the points raised in the reviews:
>
> **Weakness:**
> Presentation. Focusing solely on open models as being error-prone unnecessarily limits the scope and potential impact of the paper's contributions when it could be relevant to any base LLM.
>
> **Response:**
>
> Thank you for pointing this out. We agree that this approach could potentially extend beyond the use of open models. Meanwhile, we do wish to emphasize that focusing on open models is particularly valuable, as they are often preferred for many applications due to their cost-effectiveness, customizability, and efficiency in inference.
>
> **Weakness:** Quality. There was a limited diversity of baselines. The domain specific language looks very similar to the code-as-policies test environments. In the experimental section, the authors should contextualize the results by either explaining the best analogy to code-as-policies that they run or by explicitly discussing a code-as-policies style baseline. The authors could also try simpler variants of their method: for example, taking some data points generated by robosim rejection sampling process and putting them in the prompt instead of doing model fine-tuning and alignment.
>
> **Response:**
>
> The problem setting this work addresses is different from Code-as-Policies. While Code-as-Policies focuses on generating low-level robot actions, our work emphasizes generating high-level robot plans. The key distinction lies in evaluation: for high-level plans, the sequence of actions is critical to task success. For instance, in the task "bring me a marker from the classroom with the most markers," the robot must first visit each classroom to identify the one with the most markers and then retrieve a marker from that specific classroom. Simply bringing back any marker would not satisfy the task. In contrast, Code-as-Policies focuses solely on the final outcome, such as whether the robot successfully retrieved a marker.
>
> **Weakness:** Significance. The APIs in the roboeval benchmark are very high-level and I wouldn't be surprised if there are many solid engineering approaches to getting a performant policy that works with high-level APIs. It's unclear how well roboinstruct will scale with more complex APIs and tasks, which limits the significance of the work.
>
> **Response:**
>
> We would like to emphasize that planning with high-level actions is a widely relevant and significant area in robotics [1, 2], especially for enabling robots to execute long-horizon, high-level tasks. Moreover, RoboInstruct is not confined to the current APIs. As demonstrated in Appendix A.6, the framework can be extended to other application domains.
>
> [1] Wenlong Huang, et al. “Language Models as Zero-Shot Planners: Extracting Actionable Knowledge for Embodied Agents.” Proceedings of the 39th International Conference on Machine Learning.
>
> [2] Bo Liu, et al. “LLM+P: Empowering Large Language Models with Optimal Planning Proficiency.” CoRR, abs/2304.11477.
>
> **Question:** Could you please fix the formatting of table 5 to remove the overlapping text?
>
> **Response:**
>
> It seems there may have been a misunderstanding, as this paper does not have a Table 5. Could you clarify if this comment refers to another table or figure in our paper?

---

### Meta-Review · Area_Chair_QP5t · 2024-12-18

**Metareview:**

This paper proposes a framework to fine-tune a code LLM for robotics tasks. The paper is well written and motivated. However, at the same time the reviewers have raised several concerns, particularly on the novelty of ideas in the paper. The evaluated tasks and the real world deployments seem to be simplistic and hence are quite limited in its current form from having significant impact in the ML community.

**Additional Comments On Reviewer Discussion:**

Not much discussion as the strengths and weaknesses of this paper are quite clear.

---

### Decision · Program_Chairs · 2025-01-22

Reject